# Improving the Language Understanding Capabilities of Large Language Models Using Reinforcement Learning

## Abstract

Large language models (LLMs), primarily built on decoder-only transformer architectures, excel in natural language generation tasks and have shown promise in adapting to diverse downstream tasks using zero-shot and few-shot prompting techniques. However, these prompting methods often fall short on natural language understanding (NLU) tasks, where smaller encoder-only models like BERT-base consistently outperform LLMs on benchmarks such as GLUE and SuperGLUE. In this paper, we explore two approaches—supervised fine-tuning and proximal policy optimization (PPO)—to enhance the NLU capabilities of LLMs. To reduce the computational cost of full-model fine-tuning, we integrate low-rank adaptation (LoRA) layers, restricting updates to these layers during both supervised fine-tuning and PPO stages. In the supervised fine-tuning approach, task-specific prompts are concatenated with input queries and ground-truth labels from the NLU training corpus, optimizing the model using the next-token prediction objective. Despite this, LLMs still underperform compared to encoder-only models like BERT-base on several NLU tasks. To address this gap, we employ PPO, a reinforcement learning technique that treats each token generation as an action and evaluates the sequence of generated tokens using a reward function based on their alignment with ground-truth answers. PPO then updates the model to maximize these rewards, effectively aligning its outputs with the correct labels. Our experiments with the LLAMA2-7B-chat-hf model demonstrate that PPO-based fine-tuning significantly improves performance, delivering an average gain of 6.3 points over supervised fine-tuning on the GLUE benchmark. PPO surpasses zero-shot prompting by 38.7 points and few-shot prompting by 26.1 points on GLUE, while also outperforming these baselines by 28.8 and 28.5 points on SuperGLUE. Additionally, PPO exceeds the performance of BERT-large, a strong baseline, with an average improvement of 2.7 points on GLUE and 9.3 points on SuperGLUE. These improvements are consistent across models such as Qwen2.5-7B-Instruct and MPT-7B-chat, highlighting PPO's robustness and effectiveness in improving the NLU capabilities of LLMs. Furthermore, LLAMA2-7B-chat-hf and LLAMA2-13B-chat-hf models fine-tuned with PPO on a single dataset exhibit strong zero-shot generalization across diverse unseen datasets. On average, they outperform GPT-4o by over 4% on sentiment analysis and natural language inference tasks, achieving notable gains of 7.3% on the Mental Health dataset and more than 10.9% on SIGA-nli. Our code is publicly available at `https://anonymous.4open.science/r/LLM_NLU-BE83`.

## 1 Introduction

Large language models (LLMs) (Radford et al., 2019; Brown, 2020; Touvron et al., 2023b) have revolutionized natural language processing (NLP) with their powerful text generation capabilities, driven by their decoder-only transformer architecture (Radford, 2018). Pretrained on large amounts of unlabeled text, LLMs can generate coherent and contextually relevant content. Using prompt-based strategies like zero-shot and few-shot prompting (Brown, 2020), LLMs can tackle various downstream tasks without requiring task-specific fine-tuning. However, these methods often underperform on natural language understanding (NLU) tasks compared to encoder-only models like BERT (Devlin, 2018), which consistently excel on benchmarks such as

GLUE (Wang et al., 2019) and SuperGLUE (Wang et al., 2020). For instance, our evaluations on LLAMA2-7B-chat-hf showed that zero-shot prompting with task-specific prompts yielded an average performance of 46.1 across all GLUE datasets, while few-shot prompting improved performance to 58.7, both of which significantly lag behind BERT-base's 79.6 as shown in Table 1. This underperformance is largely due to LLMs' inability to capture bidirectional context and perform deeper semantic analysis. Improving the NLU performance of LLMs remains a challenge, as their autoregressive nature limits their ability to model the bidirectional dependencies crucial for NLU tasks (Radford et al., 2019; Brown, 2020).

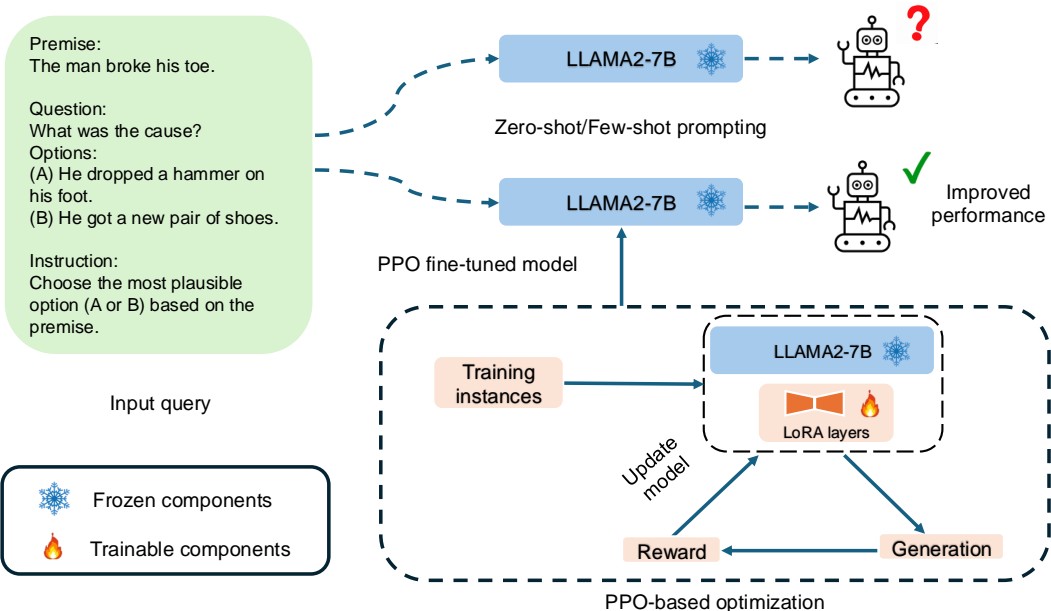

Figure 1: PPO-based fine-tuning of LLAMA2-7B-chat-hf to improve the performance on NLU tasks.

To enhance the performance of LLMs on NLU tasks, we explore two approaches. First, we apply supervised fine-tuning (SFT) of LLMs on NLU training datasets. The model is fine-tuned on input sequences consisting of task-specific prompts, training examples, and their corresponding ground-truth labels, using the next-token prediction objective. To reduce computational overhead, we employ low-rank adaptation (LoRA) layers (Hu et al., 2021a), ensuring that only these lightweight matrices are updated during fine-tuning, rather than the entire model. However, in our experiments with LLAMA2-7B-chat-hf, this approach underperforms compared to BERT-base on several GLUE datasets, including QQP, SST-2, STS-B, and MRPC, as detailed in Table 1. On average, across all GLUE datasets, BERT-base achieves a score of 79.6, outperforming LLAMA2-7B-chat-hf fine-tuned model, which attains 78.5. This indicates the need for an alternative approach to further boost performance.

To further enhance the performance of LLMs on NLU tasks, we adopt a proximal policy optimization (PPO) (Schulman et al., 2017a) based fine-tuning approach, leveraging LoRA layers to reduce computational complexity. Previous works, including A3C (Mnih et al., 2016), AlphaGo (Silver et al., 2017b), OpenAI Five (OpenAI et al., 2019), and AlphaZero (Silver et al., 2017a), have demonstrated that policy-based reinforcement learning can effectively train neural networks to perform actions in complex environments. These methods have also been widely applied to align LLM responses with human preferences (Bai et al., 2022a; Ouyang et al., 2022a) and improve reasoning capabilities (Havrilla et al., 2024). Building on this foundation, we employ PPO to improve LLM performance on NLU tasks. We frame the task of generating responses by LLMs as a reinforcement learning problem, where the sequence of input tokens represents the state $s_t$, and the token generated at each timestep $t$ is treated as the action $a_t$. After the entire sequence is generated, a heuristic-based process extracts the answer, which is compared to the ground truth label, and a reward $R$ is assigned accordingly. Our major contributions are:

- We utilize a PPO-based fine-tuning approach to improve the NLU capabilities of LLMs. To reduce computational complexity, we fine-tune only the LoRA layers.

- Our evaluation on the GLUE and SuperGLUE benchmarks using the LLAMA2-7B-chat-hf model(Touvron et al., 2023a) shows that PPO-based fine-tuning significantly outperforms zero-shot and few-shot baselines, with an average improvement of 38.7 and 26.1 points on the GLUE benchmark, and 28.8 and 28.5 points on the SuperGLUE benchmark, respectively. Additionally, PPO-based fine-tuning achieves an average gain of 6.3 points over SFT on the GLUE benchmark and outperforms BERT-large, with a 2.7-point gain on GLUE and a 9.3-point improvement on SuperGLUE benchmark.

- We evaluate the zero-shot generalization capabilities of LLAMA2-7B-chat-hf and LLAMA2-13B-chat-hf models fine-tuned using PPO. The models are fine-tuned on SST-2 for sentiment analysis tasks and MNLI for natural language inference tasks, then tested on multiple unseen datasets spanning various tasks. Both models demonstrate significant improvements over GPT-4o. LLAMA2-13B-chat-hf achieves a 4.5% improvement on Labelled Financial News, 6.2% on Babi-nli, and over 10.9% on SIGA-nli, while LLAMA2-7B-chat-hf shows a 7.3% improvement on the Mental Health dataset. On average, PPO fine-tuned LLAMA2-chat-hf models outperform GPT-4o by more than 4% on sentiment analysis and natural language inference tasks. These results underscore the effectiveness of PPO fine-tuning in improving zero-shot generalization, even compared to a significantly larger model like GPT-4o.

- The results are consistent with other LLMs such as Qwen2.5-7B-Instruct and MPT-7B-chat, demonstrating the robustness of our approach.

## 2 Related Works

### 2.1 Natural Language Understanding

Natural language understanding (NLU) tasks are crucial for evaluating a model's ability to comprehend and process human language in various contexts, such as classification, inference, and reasoning. The GLUE benchmark (Wang et al., 2019) serves as a key standard for NLU performance, covering tasks like CoLA, SST-2, MRPC, MNLI, and so on, which assess grammatical acceptability, sentiment analysis, paraphrase detection, and textual entailment. For more complex challenges, the SuperGLUE benchmark (Wang et al., 2020) was introduced, featuring more difficult tasks that require advanced reasoning and comprehension. Together, GLUE and SuperGLUE provide a comprehensive assessment of a model's language understanding capabilities.

Models such as BERT (Devlin, 2018), which utilize a bidirectional encoder architecture, have achieved state-of-the-art performance in NLU tasks. BERT's architecture allows it to capture bidirectional context. Its pretraining strategy, which uses masked language modeling (MLM), helps the model learn deep semantic representations. This combination makes BERT highly effective across a wide range of NLU tasks. The success of encoder-only models in benchmarks such as GLUE and SuperGLUE can largely be attributed to their ability to capture rich bidirectional context during pretraining, which is critical for NLU tasks.

In contrast, LLMs like GPT-2 (Radford et al., 2019), GPT-3 (Brown, 2020), and LLAMA (Touvron et al., 2023b) rely on scaling model size with decoder-only architectures, achieving significant success in text generation tasks. However, their zero-shot performance with task-specific prompts remains suboptimal on NLU tasks, such as those in the GLUE benchmark. This underperformance is attributed to their autoregressive nature, which limits their ability to capture the bidirectional dependencies crucial for deep contextual understanding (Devlin, 2018; Radford et al., 2019; Brown, 2020). Efforts to adapt LLMs for NLU have focused on prompt-based methods like few-shot prompting (Brown, 2020), which show promise but still fall short of the performance achieved by encoder-only models like BERT on these tasks.

## 2.2 Policy-based Reinforcement Learning

Policy-based reinforcement learning (RL) directly optimizes an agent's policy by learning its parameters to maximize long-term rewards. Unlike value-based methods like Q-learning (Watkins & Dayan, 1992) and DQN (Hester et al., 2018), which indirectly derive policies through value functions, policy-based methods represent the policy as a parameterized function. This function, $p_\theta(a|s)$, defines the probability of taking action $a$ in state $s$, where $\theta$ represents the policy parameters. The goal is to learn optimal parameters $\theta^*$ that maximize the expected cumulative reward, typically through policy gradient methods (Sutton et al., 1999). These methods excel in high-dimensional or continuous action spaces, where value-based methods can struggle (Deisenroth et al., 2013).

Policy-based methods in reinforcement learning (RL) have evolved significantly over time, starting with REINFORCE (Williams, 1992), which optimizes policies using the policy gradient theorem but suffers from high variance due to its reliance on Monte Carlo estimates of the reward. Monte Carlo estimates refer to calculating the total reward based on full episodes of interaction, meaning updates are made only after an entire sequence of actions and rewards is observed, which can lead to noisy and slow learning. To address this, actor-critic methods like A2C and A3C (Mnih, 2016) introduced a critic that estimates the value of the current state, allowing for smoother updates by reducing the variability in policy updates and speeding up convergence. However, these methods still faced instability when large updates caused the new policy to diverge too far from the previous one. Trust Region Policy Optimization (TRPO) (Schulman, 2015) tackled this by limiting the size of policy updates using a KL divergence constraint, but its implementation was complex and computationally expensive. Proximal policy optimization (PPO) (Schulman et al., 2017a) simplified this process by introducing a clipped objective function that keeps policy updates within a stable range while being easier to implement. PPO's balance between simplicity and stability has made it a widely adopted method in modern RL research.

In NLP, PPO has been effectively used in reinforcement learning from human feedback (RLHF) to align LLM outputs with human preferences, as seen in works like InstructGPT (Ouyang et al., 2022b) and Constitutional AI (Bai et al., 2022b). These approaches treat the LLM as a policy, where model responses are actions, and human feedback serves as rewards. PPO updates the policy based on the reward model trained on human preferences. Additionally, policy-based RL methods have been applied to enhance LLM reasoning capabilities (Ziegler et al., 2019; Havrilla et al., 2024; Hu & Shu, 2023). In this work, we apply PPO to fine-tune LLMs on NLU tasks.

## 3 Preliminaries on Application of PPO for Fine-tuning LLMs

Proximal policy optimization (PPO)(Schulman et al., 2017b) is an online reinforcement learning algorithm. In this section, we describe the process to fine-tune an LLM using PPO. During training, at each timestep $t$, the LLM (policy) generates a token prediction $a_t$ (action) based on the state $s_t$, which consists of the sequence of generated tokens up to timestep $t-1$. The final generated output is evaluated in the context of the downstream task, where the environment provides feedback in the form of rewards. The model updates its parameters based on these rewards to improve its ability to generate accurate predictions over time.

PPO uses gradient ascent to optimize the following objective, aiming to maximize cumulative rewards:

$$J(\theta) = \mathbb{E}_{(s_t, a_t) \sim \pi_{\theta'}} \left[ \min \left( \frac{p_\theta(a_t|s_t)}{p_{\theta'}(a_t|s_t)} \hat{A}_t, \text{clip}\left( \frac{p_\theta(a_t|s_t)}{p_{\theta'}(a_t|s_t)}, 1-\epsilon, 1+\epsilon \right) \hat{A}_t \right) \right]$$

Here, $p_\theta(a_t|s_t)$ is the probability of taking action $a_t$ in state $s_t$ under the current policy, while $p_{\theta'}(a_t|s_t)$ represents this probability under the old policy. In PPO, the training data—specifically, the state-action pairs $(s_t, a_t)$—are sampled using the old policy $\pi_{\theta'}$ (the LLM before it is updated), rather than the new policy currently being optimized. Thus, the ratio $\frac{p_\theta(a_t|s_t)}{p_{\theta_{\text{old}}}(a_t|s_t)}$ accounts for how much the new policy has changed relative to the old policy and adjusts the likelihood of an action accordingly. This ratio is multiplied by $\hat{A}_t$, the Generalized Advantage Estimation (GAE) (Schulman et al., 2018), which measures how much *better or worse* an action $a_t$ is compared to the expected outcome under the current policy.

$$\hat{A}_t = R_t + \gamma V_{t+1} - V_t + \gamma\lambda\hat{A}_{t+1},$$

Here, $R_t + \gamma V_{t+1} - V_t$ represents the temporal difference (TD) error (Sutton, 1988). In this expression, $R_t$ is the immediate reward received after taking action $a_t$, $V_t$ is the expected reward before the action, and $\gamma V_{t+1}$ is the discounted estimate of the future reward after the action. This term reflects how the action $a_t$ performed when compared to the expected return at state $s_t$. The second term, $\gamma \lambda \hat{A}_{t+1}$, is the smoothing factor in GAE, where $\lambda$ is the trade-off parameter. This recursive estimate allows the model to incorporate future information, making the advantage estimate more stable. Smaller values of $\lambda$ emphasize on immediate rewards, while larger values capture longer-term dependencies. The discount factor $\gamma$ controls how much emphasis is placed on future rewards compared to immediate ones, with higher values of $\gamma$ giving more weight to future rewards. $V_t$, which represents the expected future reward from state $s_t$, is estimated by a *critic model*.

The clipping function $\text{clip}(\text{ratio}, 1 - \epsilon, 1 + \epsilon)$ limits the change between the current and old policy, ensuring stable updates by preventing large deviations. This helps avoid too-large policy changes that could destabilize training. In summary, PPO optimizes the policy using gradient ascent to maximize cumulative rewards while ensuring stable updates through clipping, with the GAE providing a more stable and accurate advantage estimate by incorporating future information recursively.

**Critic Model** The critic model consists of a value head, which is a multi-layer perceptron attached to the final layer of the LLM. It takes the LLMs representation of the generated token sequence up to timestep $t$ (i.e., the state $s_t$) and predicts a scalar value representing the value function $V_t$ for that state. The critic model is updated using the square of TD error, which is computed as:

$$\delta_t = (R_t + \gamma V_{t+1} - V_t)^2, \tag{1}$$

where $\delta_t$ represents the L-2 loss between the actual reward $R_t$, combined with the discounted estimate of future rewards $\gamma V_{t+1}$, and the current predicted value $V_t$ for state $s_t$. By minimizing this TD error via gradient descent, the critic model updates its value function predictions, improving alignment with the actual rewards and future outcomes. In summary, LLM uses the PPO objective to update its policy based on feedback from the critic model, while the critic model is updated to better predict the value function for future states.

## 4 Method

To enhance the performance of LLMs on NLU tasks, we adopt two distinct fine-tuning methods. The first approach involves supervised fine-tuning, where the input consists of a concatenation of the task-specific prompt, query and the ground truth answer, with the model optimized using the next-token prediction objective. The second approach utilizes PPO, framing response generation as a reinforcement learning problem. In this setup, the sequence of input tokens until timestep $t - 1$ represents the state $s_t$, and each token generated at timestep $t$ is treated as an action $a_t$. After generating the entire sequence, a heuristic-based process extracts the final answer from this generated sequence, and is compared to the ground truth. PPO is then employed to optimize the model by maximizing the cumulative reward derived from this comparison. To reduce computational complexity, we fine-tune LoRA layers instead of the full model.

### 4.1 Task-Specific Prompt Design

We detail the construction of task-specific prompts used to query the LLM for NLU tasks. Each prompt begins with a clear task description, outlining the necessary background information to guide the model in solving the task. Following this, we specify strict requirements for the output format, ensuring that the response is encapsulated within a predefined structure, specifically between '<Judgement></Judgement>' tags. This structure ensures consistency in the model's responses, facilitating easier extraction and evaluation of the results.

For example, in the CoLA task, which assesses grammatical acceptability, the prompt is structured as follows:

```
System_prompt:
    You are an assistant to analyze the linguistic properties
```

```
    of a sentence. The task is to decide the linguistic acceptability
    of a sentence. If the sentence is linguistically correct then it
    is acceptable, else it is not.
The result you give should have the following form:
    <Judgement> {Insert only "Yes" or "No" here} </Judgement>
Prompt:
    Now judge if the sentence "{sentence}" is linguistically acceptable.
Assistant:
    <Judgement>
```

The prompt starts with background information about CoLA, specifies restrictions on the output (such as labeling a sentence as acceptable or unacceptable), and concludes with a special start token, <Judgement>, to initiate the model's response generation.

## 4.2 Supervised Fine-tuning of LLM on NLU Tasks

Given an NLU training dataset, $\mathcal{D}^{(tr)} = \{(x_i, y_i)\}_{i=1}^N$, where $x_i$ represents the input text and $y_i$ the ground truth label, we fine-tune the LLM on a sequence consisting of the task-specific prompt $p$ (described in section 4.1) concatenated with the input $x_i$ and the ground truth answer $y_i$. The model is trained using the next-token prediction objective, where it predicts the next token in the sequence by conditioning on all preceding tokens. This objective trains the model to learn to predict the correct answer for the NLU task conditioned on the task-specific prompt and input.

## 4.3 Proximal Policy Optimization for LLM Fine-tuning on NLU Tasks

We utilize PPO to fine-tune the LLM on NLU tasks, following the training protocol outlined in section 3. The reward function is specifically designed for each NLU task. In this work, we use a simple reward function, where a reward is assigned at *the end of the generation* based on alignment with the ground truth labels. We use regular expression matching to extract answers from the LLMs outputs by first locating the text within the '<Judgement></Judgement>' tags. Depending on the task, we then search for task-specific keywords (such as "yes", "no", "acceptable", or "not acceptable") to identify the answer. These extracted answers are compared with the ground truth to determine the appropriate rewards.

For instance, CoLA, which is a classification task, answers are categorized as *acceptable*, *unacceptable*, or *exceptional* (incorrect format). For STS-B, which is a regression task, the extracted answer is a floating-point number between 0 and 5. The reward per generation for classification tasks is given by $R = \mathbb{1}(\hat{y} == y_i)$, where $\hat{y}$ is the model's prediction and $y$ is the ground truth. For STS-B, a regression task, the reward per generation is calculated based on how close the prediction is to the ground truth: $R = 2.5 - |\hat{y}_i - y_i|$. *Incorrectly formatted responses* are penalized with a value of -1 for classification tasks and -2.5 for regression tasks.

## 4.4 Low-Rank Adaptation

To mitigate the computational cost of full-model fine-tuning, we employ LoRA (Hu et al., 2021b) during both the supervised fine-tuning and PPO stages. Instead of updating the entire model, we restrict the updates to LoRA layers, which significantly reduces the number of trainable parameters by decomposing the weight matrices into low-rank matrices.

# 5 Experiments

## 5.1 Experimental Setup

We trained and evaluated our models on the GLUE(Wang et al., 2019) and SuperGLUE(Wang et al., 2020) benchmarks. All experiments were conducted using instruction-tuned LLAMA2-7B models(Touvron et al., 2023a)[1]. We perform both single task and multi-task fine-tuning: 1) *Single-task Fine-tuning*: For each

---

[1]https://huggingface.co/daryl149/llama-2-7b-chat-hf

subtask within GLUE and SuperGLUE, a separate task-specific LoRA module was trained independently. 2) *Multi-task Fine-tuning*: In the multi-task setting, datasets from different subtasks within each benchmark were combined, and a single LoRA module was trained to handle all tasks simultaneously.

**Hyperparameter Settings** For PPO-based fine-tuning, grid search is performed to select the batch size in 4, 8, 12, and 16 for each task. A batch size of 24 was used across all tasks during supervised fine-tuning (SFT). The PPO epoch is set to 4, meaning that each sampled batch is used for updating the model four times. The initial learning rate for all tasks was set to $9 \times 10^{-6}$. We utilized the Adafactor optimizer for PPO training and AdamW for SFT. A cosine annealing learning rate scheduler with a warmup phase was employed, where the learning rate was gradually increased during the first 10% of training steps and then reduced to one-tenth of the initial value by the end of training. We use a rank $r = 16$ for the LoRA layers. We trained both PPO and SFT models until convergence on the validation set. The best hyperparameters were selected based on performance on the validation set. The final reported results for the GLUE and SuperGLUE are from their corresponding evaluation server. For evaluation, multinomial sampling with a temperature of 1 was used to generate a single response per data sample. The model generated responses with lengths between 12 and 32 tokens, with the generation process concluding using a special identifier `"</Judgement>"`.

### 5.2 Baselines

We evaluated the performance of our approach against three baselines:

- **Encoder-only models**: We compare our results with encoder-only transformer models, specifically BERT-base (110M parameters) and BERT-large (340M parameters)(Devlin et al., 2019).

- **Zero-shot prompting:** The model is provided with task-specific prompts, as outlined in section 4.1, along with the input query. The model is required to generate predictions solely based on these prompts and the input query, without any additional task-specific fine-tuning.

- **Few-shot prompting:** In this setting, the model is provided with both the task-specific prompt and one to five labeled examples (which ever gave the best performance) from the training dataset as demonstrations. These examples are provided as reference to guide the model in generating more accurate responses for the input query. Similarly, no task-specific fine-tuning is performed.

After generating a response, we applied regular expression matching to extract the relevant answer from the model's output. We directly matched task-specific keywords (like "yes" or "no") in the generated text to identify the answer. This extracted answer was then compared to the ground truth label to evaluate the model's performance.

### 5.3 Results on GLUE Benchmark

In this section, we present our experiments on the GLUE benchmark, comparing the results with encoder-only models such as BERT(Devlin et al., 2019). We use the LLAMA2-7B-chat-hf model as the LLM for our evaluations. The baselines include zero-shot prompting and few-shot prompting. For fine-tuning methods, we compare both supervised fine-tuning and PPO across single-task and multi-task settings. The results are summarized in Table 1. From the results, we make the following observations.

**First**, we observed that zero-shot prompting of the LLAMA2-7B-chat-hf model with task-specific prompts consistently underperformed compared to the smaller BERT-base model. LLAMA2-7B-chat-hf struggled notably on simpler tasks like SST-2, which only required classifying sentiment as positive or negative. This underscores the model's weak language understanding capabilities, with zero-shot prompting proving inadequate compared to BERT-base. **Second**, few-shot prompting showed improvements over the zero-shot baseline, achieving an average score of 58.7 compared to 46.1, but it still lagged significantly behind the BERT-base model's score of 79.6. **Third**, supervised fine-tuning (SFT) using LoRA modules for each task further boosted performance, bringing it closer to BERT's level with an average score of 78.5, though still

| Models | MNLI-m | MNLI-mm | QQP | QNLI | SST-2 | CoLA |
|---|---|---|---|---|---|---|
| **BERT-base** | 84.6 | 83.4 | 71.2 | 90.5 | 93.5 | 52.1 |
| **BERT-large** | 86.7 | 85.9 | **72.1** | 92.7 | 94.9 | **60.5** |
| **LLAMA2-7B-chat-hf** | | | | | | |
| *Zero-shot prompting* | 38.3 | 39.7 | 31.3 | 58.5 | 75.7 | 18.6 |
| *Few-shot prompting* | 62.4 | 61.7 | 30.9 | 60.7 | 84.2 | 29.0 |
| *PPO-ST* | **88.8** | 88.2 | 70.5 | 93.2 | **96.4** | 59.9 |
| *SFT-ST* | 87.0 | 86.5 | 63.8 | **93.6** | 73.8 | 50.7 |
| *PPO-MT* | 88.7 | **88.3** | 67.3 | 90.2 | 94.6 | 47.7 |
| *SFT-MT* | 84.9 | 84.5 | 62.9 | 86.0 | 72.0 | 41.4 |

| Models | STS-B | MRPC | RTE | WNLI | AX | Average |
|---|---|---|---|---|---|---|
| **BERT-base** | 85.8 | 88.9 | 66.4 | / | / | 79.6 |
| **BERT-large** | 86.5 | 89.3 | 70.1 | / | / | 82.1 |
| **LLAMA2-7B-chat-hf** | | | | | | |
| *Zero-shot prompting* | 27.5 | 66.3 | 59.3 | 44.5 | 9.2 | 46.1 |
| *Few-shot prompting* | 45.5 | 80.8 | 72.9 | 51.4 | 9.2 | 58.7 |
| *PPO-ST* | 92.6 | **89.4** | 84.3 | 74.7 | **52.7** | **84.8** |
| *SFT-ST* | 84.7 | 85.8 | 80.4 | 63.7 | 45.1 | 78.5 |
| *PPO-MT* | **94.7** | 86.7 | **86.9** | 66.4 | 43.4 | 82.9 |
| *SFT-MT* | 85.5 | 82.6 | 86.2 | **76.0** | 41.2 | 76.22 |

Table 1: GLUE test results are scored by the evaluation server (GLUE benchmark). Average column indicates the averaged performance across all the datasets excluding the WNLI and AX datasets. F1 scores are reported for QQP and MRPC, Spearman correlations for STS-B, Matthew's correlations for CoLA, and accuracy scores for the other tasks. *Zero-shot prompting* refers to prompting with task-specific prompts and an input query, while *Few-shot prompting* refers to prompting with task-specific prompts, 1-5 demonstrations (chosen based on the best performance), and an input query. *PPO* stands for proximal policy optimization, and *SFT* refers to Supervised Fine-tuning. "ST" represents Single-task, while "MT" represents Multi-task. The **bolded** results indicate the best results, and the underlined results indicate the second-best results.

slightly behind BERT-base's 79.6. **Fourth**, fine-tuning with PPO delivered the best results, achieving an average score of 84.6, surpassing even BERT-large's 82.1. Moreover, zero-shot and few-shot prompting of LLAMA2-7B-chat-hf displayed a noticeable output imbalance, with a tendency to favor certain classes or values. In contrast, models fine-tuned with PPO showed no significant bias. **Fifth**, the total computational time for PPO is approximately *1.32 times* that of SFT, indicating only a marginal increase in computational costs.

Additionally, we compared the results with multi-task training, where a *single LoRA module* was trained across all datasets using both SFT and PPO to reduce time complexity. We found that SFT on individual tasks outperformed its multi-task fine-tuning counterpart. However, while PPO on multi-task training did not perform as well as PPO on single-task training, it still outperformed BERT-large in average performance, achieving a score of 82.9 compared to BERT-large's 82.1. These results demonstrate that while single-task fine-tuning yields the best performance, multi-task training with PPO can still achieve competitive results, even surpassing state-of-the-art models like BERT-large.

### 5.4 Evaluating Zero-Shot Generalization of PPO Fine-Tuned Models and Comparison with GPT-4o

We evaluate the zero-shot generalization capabilities of LLAMA2 7B and 13B models fine-tuned using PPO on a single dataset and subsequently tested across multiple other datasets (Table 2). For sentiment analysis tasks, the models were fine-tuned on SST-2 and evaluated on diverse datasets, including Financial PhraseBank (Malo et al., 2014), Labelled Financial News (Sood, 2024), Mental Health (Gaes, 2023), and Emotion (Saravia et al., 2018). Similarly, for natural language inference (NLI) tasks, the models were fine-tuned on MNLI and evaluated on Babi-nli (Weston et al., 2015) and SIGA-nli (Nizamani et al., 2024).

| Tasks | LLAMA2-7B PPO-ST | LLAMA2-13B PPO-ST | GPT-4o |
|---|---|---|---|
| **Sentiment Analysis** | | | |
| *Financial PhraseBank* | 97.2 | **97.7** | 97.5 |
| *Labelled Financial News* | 70.2 | **72.3** | 67.8 |
| *Mental Health* | **67.2** | 66.6 | 59.9 |
| *Emotion* | **78.0** | 76.4 | 77.6 |
| **Natural Language Inference** | | | |
| *Babi-nli* | 68.3 | **69.4** | 63.2 |
| *SIGA-nli* | 46.2 | **46.3** | 35.4 |
| **Average** | 71.2 (4.3↑) | **71.5** (4.6↑) | 66.9 |

Table 2: Accuracy of different models across downstream tasks. For sentiment analysis tasks, models are fine-tuned on SST-2 and zero-shot evaluated on Financial PhraseBank (Malo et al., 2014), Labelled Financial News (Sood, 2024), Mental Health (Gaes, 2023), and Emotion (Saravia et al., 2018). Similarly, for natural language inference tasks, models are fine-tuned on MNLI and zero-shot evaluated on Babi-nli (Weston et al., 2015) and SIGA-nli (Nizamani et al., 2024). PPO-ST represents fine-tuning using Proximal Policy Optimization. Gains over GPT-4o model in the average row is indicated with green arrows.

| Tasks | LLAMA2-7B PPO-ST | LLAMA2-13B PPO-ST | GPT-4o |
|---|---|---|---|
| **Sentiment Analysis** | | | |
| *Financial PhraseBank* | (96.2, 98.1) | (96.9, 98.5) | (96.6, 98.4) |
| *Labelled Financial News* | (66.1, 74.6) | (69.0, 76.6) | (63.2, 72.2) |
| *Mental Health* | (66.6, 67.7) | (66.0, 67.1) | (59.3, 60.5) |
| *Emotion* | (77.4, 78.6) | (75.8, 77.0) | (77.0, 78.2) |
| **Natural Language Inference** | | | |
| *Babi-nli* | (64.3, 71.5) | (65.1, 73.0) | (58.8, 67.6) |
| *SIGA-nli* | (39.0, 53.9) | (40.6, 53.7) | (28.5, 42.2) |

Table 3: To quantify uncertainty in our evaluations, we generate 100 predictions for each example in the dataset. The evaluation metric is then computed for each set over the entire dataset, forming a distribution of values. The 95% confidence interval is defined by the 2.5th and 97.5th percentiles of this distribution. For sentiment analysis, models fine-tuned on SST-2 are evaluated in a zero-shot setting on Financial PhraseBank, Labelled Financial News, Mental Health, and Emotion datasets. For natural language inference, models fine-tuned on MNLI are zero-shot evaluated on Babi-NLI and SIGA-NLI.

Our results demonstrate that PPO fine-tuning improves the zero-shot performance of LLAMA2-chat-hf models compared to **GPT-4o**, a strong baseline. For sentiment analysis, LLAMA2-13B-chat-hf achieves 97.7% accuracy on Financial PhraseBank, slightly outperforming GPT-4o (97.5%). On Labelled Financial News, LLAMA2-13B-chat-hf records 72.3%, exceeding GPT-4o by 4.5%. Similarly, on the Mental Health dataset, LLAMA2-7B-chat-hf achieves 67.2%, marking a notable gain of 7.3% over GPT-4o. For the Emotion dataset, LLAMA2-7B-chat-hf achieves 78.0%, with a smaller gain of 0.4%. For NLI tasks, LLAMA2-13B-chat-hf achieves 69.4% accuracy on Babi-nli, surpassing GPT-4o by 6.2%. Additionally, LLAMA2-13B-chat-hf achieves 46.3% accuracy on SIGA-nli, outperforming GPT-4o by more than 10%. On average, both 7B and 13B versions of PPO fine-tuned LLAMA2-chat-hf models demonstrate a performance gain of over 4% compared to GPT-4o, which is significantly larger in size and highly optimized.

To ensure robust comparisons, we quantify uncertainty in our evaluations by generating 100 predictions for each example in the dataset. The evaluation metric is then computed over the entire dataset for each set,

yielding a distribution of values. The 95% confidence interval is defined by the 2.5th and 97.5th percentiles of this distribution. Results are presented in Table 3.

These results demonstrate the effectiveness of simple PPO fine-tuning on a single task-specific dataset in significantly enhancing model performance on similar tasks. LLAMA2-chat-hf models fine-tuned with PPO consistently outperform GPT-4o across diverse downstream tasks, reinforcing PPO fine-tuning as a robust approach for improving the NLU capabilities of LLMs.

We measured inference time on the Financial PhraseBank dataset with a batch size of 4. The BERT-base model, with 110M parameters, required 0.035s per step, while the LLAMA2-7B model, with 7B parameters and multi-token generation, took 0.997s per step. This difference is expected given the larger model size and the need for multiple forward passes in LLAMA2-7B. While LLM inference is slower, our focus is on improving natural language understanding with PPO, which achieves strong performance gains on both in-distribution and out-of-distribution NLU and NLI tasks.

### 5.5 Evaluation of Instruction-Following in Out-of-Distribution Tasks

To assess the instruction-following capabilities of LLMs in tasks differing from their fine-tuned format, we conduct evaluations using the LLAMA2-7B-chat-hf model fine-tuned on the SST-2 dataset. Specifically, we evaluate the performance of this model on the Amazon review task, which requires generating an integer rating between 1 and 5 based on the provided textual review. Although SST-2 and Amazon reviews both involve sentiment analysis, the two tasks differ distinctly in their input-output formatting, providing a clear measure of instruction-following adaptability.

We compare three versions of the LLAMA2-7B-chat-hf model: the original non-fine-tuned model, a version fine-tuned using SFT, and another fine-tuned with PPO. The 95% confidence intervals (CI) reported here are defined by the 2.5th and 97.5th percentiles of the bootstrap distribution. Using a consistent prompt across models, we find that the PPO-fine-tuned model achieves an accuracy of 39.35% (95% CI: 38.39, 40.29), significantly outperforming the original model, which achieves 27% accuracy (95% CI: 19.00, 36.03). Conversely, the SFT-fine-tuned model demonstrates extremely poor performance, achieving less than 1% accuracy.

| Method | Accuracy | 95% CI |
|--------|----------|--------|
| Original | 27.00 | (19.00, 36.03) |
| SFT | 0.00961 | (0.00, 0.03) |
| PPO | **39.35** | (38.39, 40.29) |

Table 4: Performance of LLAMA2-7B-chat-hf on the Amazon Review dataset. Best results are highlighted in bold.

Qualitative analysis of sampled outputs reveals that the PPO-fine-tuned model reliably adheres to the instruction format and generates detailed reasoning to support its predictions. In contrast, the SFT-fine-tuned model often fails to adapt its responses to the required format, demonstrating limited generalization capabilities. PPO fine-tuning maintains proximity to the original model distribution via a clipping mechanism, thus preserving and enhancing the model's intrinsic instruction-following capabilities. In contrast, SFT fine-tuning appears to narrow the model's learned distribution to task-specific training data, negatively impacting its original instruction-following proficiency.

### 5.6 Impact of Fine-Tuning on Language Modeling Ability

We experiment with SFT and PPO to improve NLU capabilities of LLMs and observe improved performance using PPO. However, it is crucial to ensure that fine-tuning methods do not significantly degrade the models' general language generation abilities. To assess this, we directly evaluate the PPL (jel, 1977; Chelba & Jelinek, 2000) of LLAMA2-7B-chat-hf models fine-tuned on the SST-2 dataset using the WikiText-2 test

set (Merity et al., 2016), which follows a natural human-written text distribution. We compare these fine-tuned models against the original, non-fine-tuned baseline model, with the expectation that the PPL of the fine-tuned models should closely match the baseline. Our results reveal that the original LLAMA2-7B-chat-hf achieves a perplexity of 6.939. The PPO-fine-tuned model closely maintains this baseline performance with a perplexity of 6.966, indicating minimal impact on its general language modeling capabilities. In contrast, the SFT-fine-tuned model displays a notably higher perplexity of 7.384, suggesting a significant reduction in generation capabilities due to convergence toward task-specific training distributions. We conjecture that PPO's clipping mechanism effectively constrains policy updates, preventing large deviations from the reference model and thereby preserving the original language modeling capabilities of LLMs. These findings underscore PPO's effectiveness in maintaining the general language abilities of LLMs during fine-tuning.

| Method | perplexity |
|---|---|
| Original | 6.939 |
| SFT | 7.384 |
| PPO | 6.966 |

Table 5: Perplexity of LLAMA2-7B-chat-hf on the WikiText-2 test set. Lower perplexity indicates better language modeling ability.

### 5.7 Results on SuperGLUE Benchmark

| Models | BoolQ | CB | COPA | MultiRC | ReCoRD | RTE |
|---|---|---|---|---|---|---|
| **BERT-large** | 77.4 | 75.7/83.6 | 70.6 | 70.0/24.0 | **72.0/71.3** | 71.6 |
| **BERT-large++** | 79.0 | **84.7**/90.4 | 73.8 | 70.0/24.1 | **72.0/71.3** | 79.0 |
| **LLAMA2-7B-chat-hf** | | | | | | |
| *Zero-shot prompting* | 75.8 | 26.4/43.6 | 57.0 | 51.9/20.3 | 27.0/26.2 | 59.2 |
| *Few-shot prompting* | 80.2 | 49.8/66.0 | 73.4 | 46.6/15.4 | 36.3/35.3 | 72.9 |
| *PPO-ST* | **85.9** | 74.7/88.0 | **88.6** | **82.5**/50.0 | 70.6/69.9 | **84.3** |

| Models | WiC | WSC | AXb | AXg | Average |
|---|---|---|---|---|---|
| **BERT-large** | 69.5 | 64.3 | 23.0 | 97.8/51.7 | 69.0 |
| **BERT-large++** | 69.5 | 64.3 | 38.0 | **99.4**/51.4 | 71.5 |
| **LLAMA2-7B-chat-hf** | | | | | |
| *Zero-shot prompting* | 54.4 | 52.1 | 9.1 | 64.0/55.1 | 49.5 |
| *Few-shot prompting* | 54.4 | 62.3 | 9.1 | 64.0/55.1 | 54.9 |
| *PPO-ST* | **72.1** | **78.1** | **52.7** | 91.0/**79.8** | **78.3** |

Table 6: SuperGLUE test results are scored by the evaluation server (SuperGLUE benchmark). The experimental data for BERT-large and BERT-large++ are taken from the original SuperGLUE paper (Wang et al., 2020). The metrics used in the experiments are as follows: CB: F1 / Acc; MultiRC: F1 / Exact Match; ReCoRD: F1 / Exact Match; AXb: MCC; AXg: Gender parity score / Acc. For the remaining tasks not mentioned, accuracy (Acc) is reported. Average column corresponds to the averaged performance across all the datasets. For tasks with multiple evaluation metrics, we first compute the average of those metrics to obtain a single task score, which is then used in the overall average calculation. The **bolded** results indicate the best results, and the underlined results indicate the second-best results.

We fine-tuned the LLAMA2-7B-chat-hf model using PPO on the SuperGLUE dataset and compared its performance against several baselines, including BERT-large, BERT-large++, and zero-shot and few-shot prompting of LLAMA2-7B-chat-hf. The term "BERT++" refers to a BERT model fine-tuned using the

supplementary training on intermediate labeled-data tasks (STILTs) approach (Phang et al., 2018), where the model is first fine-tuned on related transfer tasks before being fine-tuned on SuperGLUE tasks. For example, MNLI from the GLUE benchmark(Wang et al., 2019) is used as an intermediate task for CB, RTE, and BoolQ(Wang et al., 2020). In contrast, our experiments with LLM did not use this method. Our models were only fine-tuned on the datasets included in the SuperGLUE benchmark.

As shown in Table 6, the PPO-tuned LLAMA2-7B-chat-hf achieved the highest average performance, surpassing all baselines. PPO demonstrated particularly strong improvements on reasoning-intensive tasks like COPA and MultiRC, where it significantly outperformed both prompting methods and encoder-only models. These results highlight the effectiveness of PPO in improving the model's capabilities, particularly for tasks requiring reasoning and contextual understanding.

It is worth noting that on MultiRC, few-shot prompting performs slightly worse than zero-shot prompting. This may be because MultiRC involves long input contexts, and incorporating multiple examples in a few-shot prompt can cause the total input length to approach or exceed the LLMs maximum context window. Even in the one-shot setting, providing an excessively long context can dilute the model's attention, potentially leading to reduced performance.

## 5.8 Performance Comparison Across Different LLMs

| Models | STS-B | COPA |
|---|---|---|
| **BERT-large** | 86.5 | 70.6 |
| **LLAMA2-7B-chat-hf** | | |
| *Zero-shot prompting* | 27.5 | 57.0 |
| *Few-shot prompting* | 45.5 | 73.4 |
| *PPO-ST* | 92.6 | 88.6 |
| **Qwen2.5-7B-Instruct** | | |
| *Zero-shot prompting* | 83.7 | 96.6 |
| *Few-shot prompting* | 87.0 | 96.0 |
| *PPO-ST* | 92.2 | 97.0 |
| **MPT-7B-chat** | | |
| *Zero-shot prompting* | 19.7 | 57.4 |
| *Few-shot prompting* | 21.7 | 57.2 |
| *PPO-ST* | 89.3 | 84.0 |

Table 7: Performance comparison of LLAMA2-7B-chat-hf, Qwen2.5-7B-Instruct(Hui et al., 2024), and MPT-7B-chat(MosaicML, 2023) models on the GLUE STS-B and SuperGLUE COPA tasks under zero-shot prompting, few-shot prompting, and PPO based fine-tuning. Results are sourced from the official GLUE benchmark and SuperGLUE benchmark evaluation servers. For STS-B, we report Spearman correlation, and for COPA, accuracy is used as the evaluation metric.

To assess the consistency of our findings across different models, we evaluated Qwen2.5-7B-Instruct and MPT-7B-chat alongside LLAMA2-7B-chat-hf on the STS-B dataset from the GLUE benchmark and the COPA dataset from the SuperGLUE benchmark. The results confirm that PPO-based fine-tuning consistently outperforms the BERT-large model, as well as the zero-shot and few-shot prompting baselines for all LLMs, highlighting its effectiveness across different LLMs. Additionally, the effect of few-shot prompting on COPA performance varies across different LLMs, indicating that different LLMs have varying capabilities to process and follow long-context instructions, which results in variable performance outcomes.

## 5.9 Reward Curve for PPO Fine-Tuning

We present the reward curve from fine-tuning LLAMA2-7B-chat-hf using PPO in a multitask setting on the GLUE dataset. Figure 2 illustrates the reward values over training iterations, offering insights into the

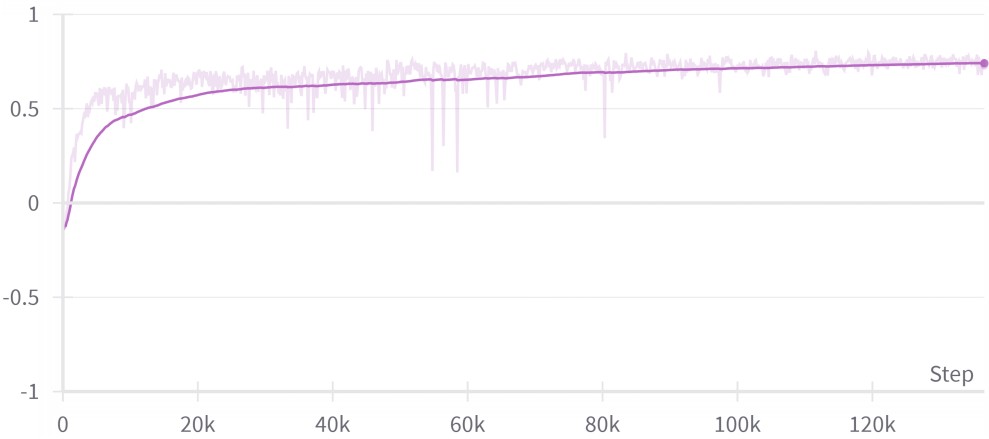

Figure 2: Reward curve for multitask PPO fine-tuning of LLAMA2-7B-chat-hf on the GLUE dataset. The plot illustrates the relationship between training iterations (x-axis) and reward values (y-axis), demonstrating the effectiveness of the PPO optimization approach in improving model performance over time.

training dynamics of the model. The curve serves as a key performance metric, tracking the model's learning progress across multiple tasks. The consistent upward trend demonstrates that PPO fine-tuning effectively improves LLAMA2-7B-chat-hf's ability to generate task-relevant outputs.

## 6 Conclusion

Prompting-based approaches, such as zero-shot and few-shot prompting, have gained popularity for adapting LLMs to downstream tasks. However, when applied to LLAMA2-7B-chat-hf, these methods underperform on NLU tasks compared to smaller encoder-only models like BERT-base and BERT-large. To address this limitation, we explore two fine-tuning strategies that leverage LoRA layers to reduce computational overhead. First, we employ supervised fine-tuning by concatenating task-specific prompts, input queries, and ground-truth labels, optimizing the model with the next-token prediction objective. While this approach improves LLAMA2-7B-chat-hf's performance over prompting-based methods, it still lags behind BERT-base on the GLUE benchmark. To further enhance performance, we adopt PPO, treating the LLM as a policy that generates the next token (action) based on the current input sequence (state). A reward function then evaluates how closely the generated tokens match the ground-truth labels, guiding updates to the policy. PPO-based fine-tuning of LLAMA2-7B-chat-hf, tested across benchmarks like GLUE and SuperGLUE, resulted in significant performance gains, outperforming strong baselines like BERT-large. Similar trends were observed in other LLMs, including Qwen2.5-7B-Instruct and MPT-7B-chat, showcasing the robustness of this approach. We also assess the zero-shot generalization capabilities of LLAMA2-7B-chat-hf and LLAMA2-13B-chat-hf models fine-tuned using PPO. By fine-tuning these models on a single dataset and testing them on multiple unseen datasets, we demonstrate their ability to generalize effectively. LLAMA2-13B-chat-hf outperforms GPT-4o with gains of 4.5% on Labelled Financial News, 6.2% on Babi-nli, and over 10.9% on SIGA-nli, while LLAMA2-7B-chat-hf achieves an improvement of 7.3% on the Mental Health dataset. These findings underscore the robustness of PPO fine-tuning in improving NLU capabilities of LLMs. Future work could extend these techniques to more diverse datasets and refine reward functions for handling complex NLU tasks.

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

# A    Evaluation on Reading Comprehension Tasks

We evaluate LLAMA2-7B-chat-hf on the SQuAD reading comprehension task, where the objective is to select a passage from a given context that best answers a question. Two training strategies are compared: Supervised Fine-Tuning (SFT), which directly uses the ground-truth answer as the training label, and Proximal Policy Optimization (PPO), which leverages reward functions based on Exact Match (EM) and F1 Score. EM metric is computed by comparing a normalized prediction against the normalized ground truth (with normalization involving lowercasing and punctuation removal); a perfect match yields an EM score of 1, otherwise 0. F1 score measures word-level overlap, balancing how many predicted words are correct (precision) and how many ground-truth words are included (recall).

Models were fine-tuned for one epoch on the SQuAD training set and evaluated on the development set. In our evaluation, zero-shot prompting yields an EM of 7.66 and an F1 score of 32.27. SFT significantly improves these metrics (EM: 59.17, F1: 76.48), while PPO further enhances performance, achieving an EM of 65.74 and an F1 score of 81.82—corresponding to improvements of 6.57 and 5.34 points over SFT, respectively.

These results indicate that optimizing with reward functions based on EM and F1 via PPO leads to further improvements in reading comprehension performance, thereby validating our approach relative to both zero-shot prompting and standard SFT.

| Method | EM | F1 |
|---|---|---|
| Original | 7.66 | 32.27 |
| SFT | 59.17 | 76.48 |
| PPO | **65.74** | **81.82** |

Table 8: Performance of LLAMA2-7B-chat-hf on the SQuAD dataset. PPO uses Exact Match and F1 as reward signals. Best results are highlighted in bold.

# B    Comparison of RL Algorithms: PPO vs. GRPO

| Algorithms | SST-2 | MRPC | RTE | CoLA | QNLI | Average | Per-Step Runtime |
|---|---|---|---|---|---|---|---|
| **SFT** | 73.8 | 85.8 | 80.4 | 50.7 | **93.6** | 76.9 | **4.124** |
| **PPO** | 96.4 | 89.4 | 84.3 | **59.9** | 93.2 | 84.6 | 4.299 |
| **GRPO** | **96.7** | **91.2** | **88.5** | 55.2 | 93.1 | **84.94** | 5.155 |

Table 9: Performance comparison of models trained with SFT, PPO, and GRPO on the GLUE SST-2, MRPC, RTE, CoLA, and QNLI tasks under zero-shot prompting. Results are sourced from the official GLUE benchmark evaluation servers. For MRPC, we report F1 score. Best results are highlighted in bold.

Our objective is to improve the natural language understanding capabilities of the base (policy) model through RL fine-tuning. In this context, we compare two approaches: PPO and Group Relative Policy Optimization (GRPO) (Shao et al., 2024). PPO is highly effective but introduces additional computational overhead. This overhead stems from the need for repeated sampling and from updating a separate critic model to compute value functions. In contrast, GRPO was designed to mitigate these costs by bypassing the critic model entirely. Instead of maintaining a separate value network, GRPO samples multiple trajectories per prompt and computes each trajectory's advantage by comparing its reward to the batch's average (and standard deviation). This method not only simplifies the architecture but also reduces memory usage.

For our experiments, we utilized the TRL library (von Werra et al., 2020) on a single Nvidia A100 GPU, with a batch size of 16 and gradient checkpointing enabled. While SFT involves a simple forward pass, loss

computation, and backward pass per step, both PPO and GRPO add extra steps such as LLM sampling, reward calculation, and advantage estimation.

As detailed in Table 9, both PPO and GRPO deliver notable performance improvements over SFT. Notably, PPO only incurs about a 4% increase in per-step runtime compared to SFT. However, GRPO's need to generate multiple responses per sample results in a higher runtime, despite its memory efficiency benefits. Overall, our analysis highlights the trade-offs between these RL algorithms: PPO offers efficient runtime with the cost of additional overhead from the critic model, while GRPO reduces memory usage at the expense of increased sampling time.

## C   Reward Function Design and Evaluation

| Method | Accuracy (%) |
|--------|--------------|
| PPO | **96.4** |
| PPO-RM | 89.7 |

(a) SST-2 performance on GLUE.

| Method | GPT Eval. Score |
|--------|-----------------|
| PPO | 3.479 |
| PPO-RM | **4.104** |

(b) Quality of generated analyses.

Table 10: Comparison of reward function designs for LLAMA2-7B-chat-hf. The model trained with a rule based reward (PPO) achieves a high SST-2 classification accuracy of 96.4%, while incorporating a sophisticated reward model (PPO-RM) significantly reduces accuracy (89.7%) but yields substantially improved analysis quality, with a GPT evaluation score of 4.104 compared to 3.479 for the simple reward. Best results are highlighted in bold.

While our primary reward function is based on matching generated outputs to true labels, we recognize that more sophisticated reward designs may be necessary for complex NLU tasks. To address this, we investigate the effect of integrating a reward model into our PPO training, with the aim of enhancing not only classification performance on SST-2 but also the quality of generated analyses.

**Reward Modeling Setup.** For the first 5,000 training samples of the SST-2 dataset, LLAMA2-7B-chat-hf generates four responses per data point. Each response includes a sentiment judgment (Positive/Negative) and a supporting analysis. To robustly rank these responses, we use GPT-4o as an evaluator. GPT-4o ranks the responses based on: (i) the correctness of the sentiment judgment (i.e., matching the ground truth), (ii) the consistency between the judgment and its accompanying analysis, and (iii) the overall factual correctness and helpfulness of the analysis. To ensure clear differentiation, we include two reference responses—one with only the correct answer and one with only the incorrect answer—and define the ranking order as: correct answer with analysis > only correct answer > incorrect answer with analysis > only incorrect answer.

**Training the Reward Model.** A reward model is then trained on this ranked dataset using a BERT-based architecture (bert-base-cased). For each input $x$, we consider pairs of responses $(y_w, y_l)$, where $y_w$ denotes a response ranked higher by our evaluator (GPT-4o) due to its correct sentiment and coherent analysis, and $y_l$ denotes a lower-ranked response. The model learns to assign higher scores to better responses via a pairwise ranking loss:

$$L(\theta) = -\mathbb{E}_{(x,y_w,y_l)\sim D}\left[\log \sigma\left(r_\theta(x, y_w) - r_\theta(x, y_l)\right)\right],$$

where $r_\theta(x, y)$ is the score assigned to response $y$ given $x$, and $\sigma$ is the sigmoid function converting the score difference into a probability. This loss encourages the reward model to output higher scores for responses with superior judgments and analyses.

**Incorporating the Reward Model into PPO Training.** During PPO training on SST-2, LLM is tasked with generating both a sentiment judgment and an analysis. The trained reward model provides the reward signal by scoring these outputs. As shown in Table 10a, while the PPO model trained with reward signals from the reward model (PPO-RM) produces analyses of higher quality, it suffers from a significant reduction in classification performance, dropping from 96.4% to 89.7%. We believe this discrepancy might be due to the limited sample size used for reward model training and potential reward hacking Amodei et al. (2016) during optimization. However, we will explore this further in our future works.

**Evaluation of Generated Analyses.** To further assess the impact of our reward design, we evaluated the quality of generated analyses. We sampled 100 data points from three models: the original LLAMA2-7B-chat-hf, the PPO model trained using only correct-answer rewards (PPO), and the PPO model trained with the reward model (PPO-RM). GPT-4o then scored each analysis on a scale from 1 to 5 based on answer correctness and logical coherence. As indicated in Table 10b, the PPO model using reward model signals achieved the highest average score, suggesting that a more complex reward function can enhance the quality of generated outputs.

In summary, while the integration of a reward model in PPO training significantly reduces classification performance compared to using only correct-answer rewards, it considerably improves the GPT evaluation scores of the analyses produced by the LLM.

