# OpenReview forum: "Improving the Language Understanding Capabilities of Large Language Models Using Reinforcement Learning"
_TMLR — Rejected by TMLR_

### Review · Reviewer_D6n7 · 2025-02-01

**Summary Of Contributions:**

This paper explores how reinforcement learning (PPO) can enhance LLMs' performance on natural language understanding tasks. The experiments are conducted using the LLAMA2-7B model, and the results are compared to those of the BERT models and GPT-4o. The results demonstrate that PPO fine-tuning effectively improves LLM performance on GLUE and SuperGLUE benchmarks while exhibiting zero-shot generalization capabilities.

**Audience:**

Yes

**Claims And Evidence:**

Yes

**Requested Changes:**

Please see the pros and cons, and consider to revise the manuscript as suggested.

**Strengths And Weaknesses:**

Strengths:
1. The use of PPO for fine-tuning LLMs, treating generated sequences as actions executed by the policy, and constructing reward functions based on ground-truth labels, enables better alignment with NLU tasks.
2. Comprehensive experiments conducted on GLUE and SuperGLUE benchmarks, covering various comparative studies including single-task/multi-task training and zero-shot/few-shot prompt learning.
3. PPO fine-tuning significantly outperforms BERT series models on multiple NLU tasks, improving SFT by 6.3 points on GLUE and exceeding GPT-4o by over 4% on certain tasks.

Weaknesses:
1. The authors mention that PPO's computation time is approximately 1.32 times that of SFT, computational cost remains a concern. PPO training requires multiple sampling and policy network updates, which may lead to substantial computational overhead when training on large-scale datasets. The authors could further explore methods to optimize the PPO training process.

2. Reward Function Design: The reward function in the paper is relatively simple, primarily based on the match between generated outputs and true labels. However, a more sophisticated reward function design may be necessary for more complex NLU tasks. A detailed discussion of the rationality and effectiveness of the reward function design, along with additional experiments to validate its performance, is needed.

3. The performance of few-shot prompts is weaker than zero-shot in tasks like SuperGLUE, and the reasons for this need to be explored. Limitations of Multi-task Training: Multi-task PPO performance is lower than single-task, but the specific reasons are not analyzed.

---

> ### Author Response · Authors · 2025-02-28
> **Author response (Part 1)**
>
> ### 1. The authors mention that PPO's computation time is approximately 1.32 times that of SFT, computational cost remains a concern. PPO training requires multiple sampling and policy network updates, which may lead to substantial computational overhead when training on large-scale datasets. The authors could further explore methods to optimize the PPO training process.
>
> We appreciate the reviewer’s concern about computational cost. In our work, we focus on updating the base model (policy model) to improve its natural language understanding capabilities. Although PPO is highly effective, it does indeed introduce additional overhead because of repeated sampling and the critic model’s value function updates.
>
> To investigate more efficient alternatives, we experimented with a newly proposed approach called Group Relative Policy Optimization (GRPO) [1], which aims to bypass the critic model. Specifically, GRPO samples multiple trajectories per prompt and computes each trajectory’s advantage by comparing its reward to the average (and standard deviation) of the entire batch of sampled trajectories. Because no separate critic model is needed, GRPO can reduce some memory usage.
>
> | Algorithms | SST-2 | MRPC | RTE  | CoLA | QNLI | Average | Per-step runtime |
> |------------|------:|-----:|-----:|-----:|-----:|--------:|--------:|
> | SFT    | 73.8  | 85.8 | 80.4 | 50.7 | 93.6 | 76.9    | 4.124   |
> | PPO    | 96.4  | 89.4 | 84.3 | 59.9 | 93.2 | 84.6    | 4.299   |
> | GRPO   | 96.7  | 91.2 | 88.5 | 55.2 | 93.1 | 84.94   | 5.155   |
>
> While GRPO yields a modest performance gain over PPO (an average score of 84.94 vs. 84.6 on these tasks), its need to generate multiple samples per prompt increases runtime (5.155 vs. 4.299). Thus, it trades memory savings for a higher time cost. We present these experimental results in Appendix B (Table 9) of the updated manuscript.
>
> In future work, we plan to explore additional optimizations—such as more efficient sampling schemes or partial critic usage—to reduce both memory and time overhead. Our ultimate goal is to preserve or improve upon PPO’s performance while mitigating the heavy computational demands that come from repeated policy updates and sampling.
>
> [1] Shao, Zhihong, Peiyi Wang, Qihao Zhu, Runxin Xu, Junxiao Song, Xiao Bi, Haowei Zhang et al. "Deepseekmath: Pushing the limits of mathematical reasoning in open language models." arXiv preprint arXiv:2402.03300 (2024).

---

> ### Author Response · Authors · 2025-02-28
> **Author response (Part 2)**
>
> ### 2. Reward Function Design: The reward function in the paper is relatively simple, primarily based on the match between generated outputs and true labels. However, a more sophisticated reward function design may be necessary for more complex NLU tasks. A detailed discussion of the rationality and effectiveness of the reward function design, along with additional experiments to validate its performance, is needed.
>
> We thank the reviewer for the suggestion. In response, we integrated a more sophisticated reward modeling framework into our PPO training to capture nuanced aspects of output quality beyond simple label matching. For the first 5,000 training samples of SST-2 dataset, LLAMA2-7B-chat-hf generates four responses per sample, each containing a sentiment judgment and an accompanying analysis.
>
> 1. We use GPT-4o to rank these responses based on judgment correctness, analysis consistency, accuracy, and helpfulness, and we include two reference responses—one with only the correct answer and one with only the incorrect answer—to ensure robust ranking.
> 2. A BERT-based reward model is then trained on this ranked dataset using a ranking loss that encourages higher scores for better responses.
> 3. During PPO training on SST-2, the model generates both a sentiment judgment and an analysis, and the trained reward model scores these outputs to provide the reward signal.
>
> Our experiments show that while the PPO model using reward model signals (PPO-RM) produces analyses of higher quality, as reflected by improved GPT-4o evaluation scores, it suffers a reduction in classification performance—from 96.4% to 89.7%. We attribute this trade-off to the limited sample size for reward model training and potential reward hacking. Nonetheless, these results demonstrate that a more nuanced reward function can capture additional aspects of output quality beyond simple label matching, motivating further exploration of advanced reward designs for complex NLU tasks. **More detailed experimental setup can be found in Appendix C (Table 10) of the updated manuscript.**
>
> | Method  | Accuracy (%) | GPT Eval. Score |
> |---------|------------|-----------------|
> | PPO     | 96.4       | 3.479           |
> | PPO-RM  | 89.7       | 4.104           |

---

> ### Author Response · Authors · 2025-02-28
> **Author response (Part 3)**
>
> ### 3. The performance of few-shot prompts is weaker than zero-shot in tasks like SuperGLUE, and the reasons for this need to be explored. Limitations of Multi-task Training: Multi-task PPO performance is lower than single-task, but the specific reasons are not analyzed.
>
> Thank you for raising these important points. Regarding few-shot prompting, we initially observed that few-shot performance on tasks such as SuperGLUE was weaker than zero-shot. Upon further investigation, we discovered that using a five-shot prompt—which includes five examples as demonstrations—sometimes exceeded the default context length of LLAMA. To address this, we adjusted the number of demonstrations (choosing between one and five per test example) to optimize performance. With this adjustment, few-shot prompting now outperforms zero-shot prompting, and we have updated our results accordingly. However, in MultiRC, using only one demonstration in the prompt still underperformed compared to zero-shot, because even a single example increases the input length significantly, which could be leading to suboptimal performance. These results are shown in Section 5.7 (Table 6) and Section 5.8 (Table 7) of the updated manuscript.
>
> Regarding multi-task PPO training, we acknowledge that multi-task models may underperform compared to single-task fine-tuned models due to task interference. We also believe that the amount of training data and the uniformity of its distribution across tasks play critical roles. For example, in GLUE, MNLI—which has the largest dataset—has its samples predominantly in the later part of the training set. As a result, the multi-task model achieves MNLI performance comparable to the single-task model, while performance on other tasks may suffer.
>
> We appreciate the reviewer’s feedback, and these insights have led us to refine both our prompting strategies and our discussion of multi-task training limitations.

---

### Review · Reviewer_58dr · 2025-02-16

**Summary Of Contributions:**

This paper explores improving natural language understanding (NLU) tasks in autoregressive, decoder-only LLMs like LLAMA2-7B using reinforcement learning (PPO) based fine-tuning, comparing it to supervised fine-tuning (SFT) and zero/few shot performance of off-the-shelf models.

Key contributions

- Fine-tuning comparison: PPO significantly outperforms SFT on GLUE and SuperGLUE, exceeding BERT-large by up to 9.3 points.
- Zero-shot generalization: PPO fine-tuned models surpass GPT-4o by 4% on average, with notable gains on sentiment analysis and inference tasks.
- Model evaluation: Results hold across LLAMA2-7B, MPT-7B, and Qwen2.5-7B, demonstrating PPO’s robustness.
- Computational efficiency: PPO is ~1.32× costlier than SFT but remains feasible with LoRA optimization.

Overall, the paper reports that PPO fine-tuning boosts LLMs' NLU performance, surpassing SFT, prompting baselines, and BERT-large on GLUE and SuperGLUE. It also enhances zero-shot generalization, outperforming GPT-4o on sentiment and inference tasks across multiple models.

**Audience:**

No

**Broader Impact Concerns:**

No such concerns

**Claims And Evidence:**

Yes

**Requested Changes:**

- Can the authors elaborate on the key take-aways from this work and the audience for these findings? I feel the core contribution of the paper, i.e., beneifts of RL-based methods for aligning base pretrained LLMs and employing LoRA, are standard practice at this point. The supervised setting of the reward model (i.e., using a labeled dataset for classification tasks) has also been explored before. Given this I struggle to find what the TMLR audience should take from this paper.

- Overall this work feels out of date. While the introduction cites recent foundational work around LLM alignment (Bai et al., 2022a; Ouyang et al., 2022a, Havrilla et al., 2024) it doesn't contextualizate these in terms of what this paper is contributing.

- Given the above, can the authors differentiate this work from other papers exploring PPO for aligning LLMs (generally or specifically for targeted classification rather than text generation tasks).

- One suggestion (I don't feel super strongly on this) -- there was a bit more useful information in the context of computational benefits of different fine-tuning methods (discussed briefly in "5.3 Results on GLUE Benchmark" with PPO vs SFT). Some concrete analysis around classification tasks and the computational benefits of using BERT vs. 7B+ LLMs would be one possible improvement.

**Strengths And Weaknesses:**

### Strengths

- Demonstrates effectiveness of PPO fine-tuning for NLU: PPO fine-tuning significantly improves LLMs' performance on NLU tasks, addressing their historical shortcomings compared to encoder-only models like BERT.
- The evaluation of models on unseen datasets (e.g., SIGA-nli, Mental Health dataset) provides some argument for PPO fine-tuning's broader applicability.
- Comparison across different fine-tuning techniques: The study provides a direct comparison between SFT and PPO, along with standard prompting baselines, which is valuable for practitioners selecting adaptation strategies for LLMs.
- The discussion on LoRA's effectiveness in reducing fine-tuning overhead is relevant given increasing concerns about the costs associated with large-scale LLM adaptation.


### Weaknesses

- I think the framing of autoregressive LLMs being poor at NLU tasks, articulated in the introduction is off base. Autoregressive LLMs that haven't been aligned perform poorly on many prompting tasks. This is a well established property of LLMs.
- The LLM backbones used in this paper (Llama-2, MPT-7B, Qwen2.5-7B) are not instruction tuned, so it's not surprising their performance for zero/few shot prompting is poor. Employing some alignment method (e.g., RL, SFT) is the standard practice for improving performance in chat/prompting/k-shot learning.
- Evaluation datasets are limited and narrowly scoped given the opening framing on NLU tasks.
- There is some risk in using older datasets with a clear risk of pretraining leakage. SuperGLUE in particular, has been the subject of research around pretraining contamination. With most experiments relying on adapting LLMs with GLUE/SuperGLUE datasets (vs newer or novel classificaiton datasets) this undercuts the rigor of the experimental setup. See the nice summary in https://arxiv.org/html/2406.14644v1
> Elazar et al. (2023) presents another comprehensive analysis that explores the overlap between pre-training corpora and the SuperGLUE Sarlin et al. (2020) benchmark. Their findings show significant contamination of several widely used pre-training corpora such as RedPajama Computer (2023), Oscar Li et al. (2020), Pile Gao et al. (2020) and C4 Raffel et al. (2023), instances of contamination in some SuperGLUE datasets reaching as high as 100%.

- No statistical tests (e.g., test set bootstrapping) to report results (mean + confidence intervals) or compare models

---

> ### Author Response · Authors · 2025-02-28
> **Author response (Part 1)**
>
> ### 1. I think the framing of autoregressive LLMs being poor at NLU tasks, articulated in the introduction, is off base. Autoregressive LLMs that haven't been aligned perform poorly on many prompting tasks. This is a well established property of LLMs.
>
> We thank the reviewer for this important observation. We would like to clarify that all the models used in our work are indeed instruction fine-tuned and human preference aligned. Specifically, we employed the following variants: Llama2-7B (llama-2-7b-chat-hf), Qwen2.5-7B (Qwen2.5-7B-Instruct), MPT-7B (mpt-7b-chat), and Llama2-13B (llama-2-13b-chat-hf). We will make this point clearer in the manuscript. Notably, even these instruction-tuned models still underperform on NLU tasks compared to smaller encoder-based models like BERT. Our experiments show that fine-tuning these LLMs using PPO on a reward model specifically designed for NLU tasks significantly enhances their performance, as evidenced by the improvements observed across various baselines.
>
> ### 2. The LLM backbones used in this paper (Llama-2, MPT-7B, Qwen2.5-7B) are not instruction tuned, so it's not surprising their performance for zero/few shot prompting is poor. Employing some alignment method (e.g., RL, SFT) is the standard practice for improving performance in chat/prompting/k-shot learning.
>
> We appreciate the reviewer’s comment and would like to clarify that the models used in our study are indeed instruction-tuned. Specifically, our experiments utilize Llama2-7B (llama-2-7b-chat-hf), Qwen2.5-7B (Qwen2.5-7B-Instruct), MPT-7B (mpt-7b-chat), and Llama2-13B (llama-2-13b-chat-hf). We will make this explicit in the revised manuscript. Nonetheless, we find it noteworthy that even with instruction tuning, the zero-shot and few-shot performance of these models on NLU tasks remains inferior to that of smaller encoder-based models like BERT—a surprising result given their extensive pretraining and instruction fine-tuning. To address this issue, we utilize a PPO fine-tuning approach using a reward model specifically designed for NLU, which has been shown to outperform standard SFT methods.

---

> ### Author Response · Authors · 2025-02-28
> **Author response (Part 2)**
>
> ### 3. Evaluation datasets are limited and narrowly scoped given the opening framing on NLU tasks.
>
>
> To address the reviewer’s concern regarding dataset scope, we expanded our evaluation to reading comprehension by including the SQuAD dataset. We fine-tuned models for one epoch on SQuAD’s training set and evaluated on the development set, comparing SFT (using ground-truth answers as labels) with PPO (using exact match and word level F1 scores as rewards:
> | Method   | Exact Match | F1    |
> |----------|------------|-------|
> | Original | 7.66       | 32.27 |
> | SFT      | 59.17      | 76.48 |
> | PPO      | 65.74      | 81.82 |
>
> Exact Match checks if the predicted answer exactly matches the ground truth after basic normalization such as lower casing and removing punctuations, while F1 measures word-level overlap, balancing how many predicted words are correct (precision) and how many ground-truth words are included (recall). PPO improves Exact Match on the evaluation set by 6.57 and F1 by 5.34 over SFT, demonstrating the effectiveness of the proposed method. This part of the experiment is presented in Appendix A, and the experimental results are shown in Table 8 of the updated manuscript.
>
>
> We also conducted extensive NLU evaluations on GLUE and SuperGLUE using LLAMA2-7B (-chat-hf version), where PPO-based fine-tuning substantially outperforms zero-shot and few-shot baselines by 38.7 and 26.1 points on GLUE and by 28.8 and 28.5 points on SuperGLUE, respectively. Additionally, PPO-based fine-tuning exceeds SFT by 6.3 points on GLUE and outperforms BERT-large by 2.7 points on GLUE and 9.3 points on SuperGLUE. The experimental results are presented in Section 5.3 (Table 1) and Section 5.7 (Table 6) of the updated manuscript.
>
> For zero-shot generalization, we first fine-tuned LLAMA2-7B and LLAMA2-13B (-chat-hf versions) on SST-2 for sentiment analysis and MNLI for NLI tasks. We then evaluated them on unseen sentiment and NLI tasks, namely Financial PhraseBank, Labelled Financial News, Mental Health, Emotion, Babi-NLI, and SIGA-NLI, where they consistently outperformed GPT-4o, achieving improvements of up to 10.9%. The results are detailed in Section 5.4 (Table 2 and Table 3) of the updated manuscript.
>
> To further validate our approach, we conducted additional experiments using Qwen2.5-7B (-Instruct version) and MPT-7B (-chat version), observing similar performance improvements. These results are presented in Section 5.8 (Table 7) of the updated manuscript. Our findings confirm that PPO-based fine-tuning is both robust and effective across a diverse range of datasets covering Sentiment Analysis & Emotion Detection, Natural Language Inference (NLI), Paraphrase & Semantic Similarity, Grammaticality & Linguistic Acceptability, Question Answering & Reading Comprehension, Word Sense Disambiguation, and Commonsense & Reasoning Challenges in NLU.

---

> > ### Author Response · Authors · 2025-02-28
> > **Author response (Part 3)**
> >
> > ### 4. Risk of dataset contamination in older benchmarks like SuperGLUE.
> >
> > We appreciate the reviewer’s concerns regarding dataset contamination, especially with SuperGLUE. We would like to clarify that our evaluations are conducted using the official online test sets on their respective evaluation servers for both GLUE and SuperGLUE, where test labels remain unreleased. This setup guarantees that our models have no access to the ground-truth labels during training, effectively eliminating any risk of label leakage. In fact, the performance of zero-shot prompting of LLAMA-2 7B (-chat-hf version) on SuperGLUE is notably inferior to that of BERT-large, underscoring that our models are not benefiting from any such leakage. Furthermore, to ensure that our RL fine-tuning genuinely improves model generalization—and does not merely exploit distributional artifacts present in older datasets—we carried out zero-shot evaluations on benchmarks published after LLAMA-2’s pretraining, such as BabiNLI and SIGA-nli. Specifically, we fine-tuned LLAMA-2 7B and 13B (-chat-hf versions) using PPO on the MNLI dataset and directly evaluated them on these newer, out-of-distribution tasks. The significant performance gains observed on these datasets confirm that our PPO fine-tuning enhances LLM performance on NLI tasks by improving generalization rather than simply memorizing pre-training data artifacts. Together, these experimental approaches ensure that our reported improvements reflect genuine advances in model performance and reasoning capabilities.
> >
> > &nbsp;
> >
> > ### 5. Lack of statistical tests (e.g., bootstrapping) to report confidence intervals and compare models.
> >
> > We thank the reviewer for the suggestion regarding statistical tests and confidence intervals. Our GLUE and SuperGLUE results come from the official online evaluation servers, where test set labels are hidden; therefore, applying bootstrapping to these benchmarks is not feasible.
> >
> > For our downstream tasks—Financial PhraseBank, Labelled Financial News, Mental Health, Emotion, Babi-nli, and SIGA-nli—we compute 95% confidence intervals using the following bootstrapping procedure: For each example in the dataset, we generate 100 predictions. The evaluation metric is then computed over the entire dataset for each set of predictions, producing a distribution of metric values. The 2.5th and 97.5th percentiles of this distribution are used as the bounds of our 95% confidence interval. We report the computed 95% confidence intervals in Section 5.4 (Table 3) of the updated manuscript.
> >
> > | Tasks | LLAMA2-7B PPO-ST | LLAMA2-13B PPO-ST | GPT-4o |
> > |-----------|----------------------|----------------------|------------|
> > | **Sentiment Analysis** | | | |
> > | Financial PhraseBank | (96.2, 98.1) | (96.9, 98.5) | (96.6, 98.4) |
> > | Labelled Financial News | (66.1, 74.6) | (69.0, 76.6) | (63.2, 72.2) |
> > | Mental Health | (66.6, 67.7) | (66.0, 67.1) | (59.3, 60.5) |
> > | Emotion | (77.4, 78.6) | (75.8, 77.0) | (77.0, 78.2) |
> > | **Natural Language Inference** | | | |
> > | Babi-nli | (64.3, 71.5) | (65.1, 73.0) | (58.8, 67.6) |
> > | SIGA-nli | (39.0, 53.9) | (40.6, 53.7) | (28.5, 42.2) |

---

> ### Author Response · Authors · 2025-02-28
> **Author response (Part 4)**
>
> ### 6. Response to weaknesses (Points 1, 2, and 3) – Key contributions and relevance to the TMLR audience
>
> We thank the reviewer for highlighting important works on LLM alignment for human preference, harmless content generation, and improved reasoning ability [1,2,3]. However, our work focuses on improving the NLU capabilities of widely used, instruction fine-tuned, and human preference-aligned models—specifically, Llama2-7B (Llama-2-7b-chat-hf), Qwen2.5-7B (Qwen2.5-7B-Instruct), MPT-7B (MPT-7b-chat), and Llama2-13B (Llama-2-13b-chat-hf). Despite their alignment, these models exhibit surprisingly poor zero-shot and few-shot performance on many NLU tasks, often lagging behind smaller encoder-only models like BERT. For instance, their performance on relatively simple tasks such as SST-2 remains suboptimal.
>
> Our study aims to enhance these LLMs' NLU performance across a diverse set of tasks, including Sentiment Analysis & Emotion Detection, Natural Language Inference (NLI), Paraphrase & Semantic Similarity, Grammaticality & Linguistic Acceptability, Question Answering & Reading Comprehension, Word Sense Disambiguation, and Commonsense & Reasoning Challenges in NLU. While standard SFT on the training set yields only marginal improvements, we observed that employing PPO with an NLU-based reward model leads to significant performance gains, both in-domain and on unseen out-of-domain tasks.
>
> To assess generalization, we fine-tuned these models with PPO on SST-2 for sentiment analysis and MNLI for natural language inference, then evaluated them zero-shot on unseen datasets. For sentiment analysis, we tested on Financial PhraseBank, Labelled Financial News, Mental Health, and Emotion datasets, while for NLI, we evaluated on BabiNLI and SIGA-nli. Our results demonstrate substantial improvements over other baselines. The observed gains suggest that PPO-based fine-tuning not only enhances performance on in-domain tasks but also strengthens overall NLU capabilities, improving transfer to diverse, out-of-distribution datasets.
>
> Additionally, we compared our fine-tuned models against GPT-4o—a significantly larger and highly optimized model. Our PPO fine-tuned Llama2 models consistently outperformed GPT-4o by more than 4% on average across sentiment analysis and NLI tasks. Notably, Llama2-13B achieved a 4.5% improvement on Labelled Financial News, 6.2% on BabiNLI, and 10.9% on SIGA-nli over GPT-4o, while Llama2-7B demonstrated a 7.3% improvement on the Mental Health dataset over GPT-4o.
>
> We further extended our analysis to the SQuAD reading comprehension task, where similar improvements were observed. These results provide empirical evidence that PPO fine-tuning serves as an effective approach for enhancing NLU performance beyond standard SFT, offering new insights into optimizing instruction-aligned models for downstream NLU tasks.
>
> [1] Bai, Yuntao, Andy Jones, Kamal Ndousse, Amanda Askell, Anna Chen, Nova DasSarma, Dawn Drain et al. "Training a helpful and harmless assistant with reinforcement learning from human feedback." arXiv preprint arXiv:2204.05862(2022).
>
> [2] Havrilla, Alex, Yuqing Du, Sharath Chandra Raparthy, Christoforos Nalmpantis, Jane Dwivedi-Yu, Maksym Zhuravinskyi, Eric Hambro, Sainbayar Sukhbaatar, and Roberta Raileanu. "Teaching large language models to reason with reinforcement learning." arXiv preprint arXiv:2403.04642(2024).
>
> [3] Ouyang, Long, Jeffrey Wu, Xu Jiang, Diogo Almeida, Carroll Wainwright, Pamela Mishkin, Chong Zhang et al. "Training language models to follow instructions with human feedback." Advances in neural information processing systems 35 (2022): 27730-27744.

---

> ### Author Response · Authors · 2025-02-28
> **Author response (Part 5)**
>
> ### 7. One suggestion (I don't feel super strongly on this) -- there was a bit more useful information in the context of computational benefits of different fine-tuning methods (discussed briefly in "5.3 Results on GLUE Benchmark" with PPO vs SFT). Some concrete analysis around classification tasks and the computational benefits of using BERT vs. 7B+ LLMs would be one possible improvement.
>
> We thank the reviewer for the insightful suggestion. In the revised manuscript, detailed in Appendix B, Table 9, we report the computational costs for both SFT and PPO. Our results indicate that PPO’s per-step runtime is roughly 4% higher than that of SFT, and its overall training time is about 1.32 times longer, a modest increase justified by the performance gains achieved with PPO.
>
> We measured inference time on the Financial PhraseBank dataset with a batch size of 4. The BERT-base model, with 110M parameters, required 0.035s per step, while the LLaMA2-7B model, with 7B parameters and multi-token generation, took 0.997s per step. This difference is expected given the larger model size and the need for multiple forward passes in LLaMA2-7B. While LLM inference is slower, our focus is on improving natural language understanding with PPO, which achieves strong performance gains on both in-distribution and out-of-distribution NLU and NLI tasks. We included this detail in section 5.4 in the updated manuscript.

---

### Review · Reviewer_u7ci · 2025-03-07

**Summary Of Contributions:**

The authors argue that the poor performance of LLMs on NLU tasks is largely due to their inability to capture bidirectional context and perform deeper semantic analysis. Therefore, this paper enhances the NLU capabilities of LLMs using two methods— SFT and PPO—while reducing computational costs through LoRA layers. The experiments demonstrate that models fine-tuned with PPO significantly outperform SFT and other baselines, such as zero-shot and few-shot prompting methods, on multiple NLU tasks.

**Audience:**

Yes

**Claims And Evidence:**

Yes

**Requested Changes:**

Overall, my impression of the current version is that the paper lacks sufficient research motivation and innovation. The methods used are relatively superficial, and the authors do not provide enough thoughtful insights. Therefore, I believe the authors should reconsider the issues raised in the "Weaknesses" section and revise the manuscript accordingly.

**Strengths And Weaknesses:**

## Strengths:
**1.Comprehensive Experiments:** The paper conducts extensive experiments on the GLUE and SuperGLUE benchmarks using the LLAMA2-7B-chat-hf model, comparing it with various baselines and evaluating its performance under different settings (e.g., single-task and multi-task fine-tuning).
**2.Effective Methods:** The methods used in the paper indeed improve the NLU capabilities of the models.

## Weaknesses:
**1.Lack of Innovation:** Applying SFT and PPO to enhance LLMs' NLU performance has already been explored in prior research. The paper does not propose new frameworks or algorithms but rather repeats existing methods.
**2.Inaccurate Experimental Approach:** The authors claim that LLMs' poor performance on NLU tasks is due to their inability to capture bidirectional context and perform deeper semantic analysis. However, the rationale for choosing SFT and PPO to address this issue is insufficient. The authors fail to clarify why SFT and PPO enable the model to capture context and perform deep semantic analysis.
**3.Insufficient Analysis of Results:** The paper merely describes the experimental results without providing a detailed theoretical explanation. For example, why does SFT underperform compared to PPO? How does PPO influence the model's parameter updates and semantic representations?

---

> ### Author Response · Authors · 2025-03-17
> **Author response (Part 1)**
>
> ### 1. Lack of Innovation: Applying SFT and PPO to enhance LLMs' NLU performance has already been explored in prior research. The paper does not propose new frameworks or algorithms but rather repeats existing methods.
>
> We appreciate the reviewer’s comment and the opportunity to clarify the novelty of our approach. While prior works [1] and [2] have indeed explored the use of supervised fine-tuning (SFT) and reinforcement learning from human feedback (RLHF) methods to enhance LLM performance on natural language understanding (NLU) tasks, these studies rely on extensive synthetic data generation or sophisticated reward models leveraging powerful LLMs such as GPT-4.
>
> Specifically, [1] addresses NLU performance by generating large-scale synthetic datasets for further SFT training, while [2] augments existing NLU training datasets with rationales provided by GPT-4 and employs RLHF with complex reward models. *Unlike these approaches, our method introduces a simpler yet effective framework: directly applying proximal policy optimization (PPO) fine-tuning on existing, standard NLU training datasets, with a straightforward heuristic reward function (assigning a reward of 1 for correct answers and 0 for incorrect ones).*
>
> Despite its simplicity, our approach yields substantial improvements: PPO-based fine-tuning significantly outperforms zero-shot and few-shot prompting methods, SFT fine-tuning on NLU datasets, and even surpasses advanced models like GPT-4o. Additionally, our rigorous empirical evaluation, which covers a broad spectrum of NLU tasks—including Sentiment Analysis & Emotion Detection, Natural Language Inference, Paraphrase & Semantic Similarity, Grammaticality & Linguistic Acceptability, Question Answering & Reading Comprehension, Word Sense Disambiguation, and Commonsense & Reasoning—demonstrates superior generalization and robustness.
> Thus, our primary innovation lies not only in showcasing the effectiveness of PPO fine-tuning (over SFT) without relying on additional synthetic data or sophisticated reward models but also in systematically and rigorously demonstrating its efficacy and generalizability across a diverse set of NLU tasks and evaluation scenarios, which has not been comprehensively explored in previous literature.
>
> [1] Yuan, Lin, Jun Xu, Honghao Gui, Mengshu Sun, Zhiqiang Zhang, Lei Liang, and Jun Zhou. "Improving Natural Language Understanding for LLMs via Large-Scale Instruction Synthesis." arXiv preprint arXiv:2502.03843 (2025).
>
> [2] Liao, Kuo, Shuang Li, Meng Zhao, Liqun Liu, Mengge Xue, Zhenyu Hu, Honglin Han, and Chengguo Yin. "Enhancing Reinforcement Learning with Label-Sensitive Reward for Natural Language Understanding." arXiv preprint arXiv:2405.19763 (2024).

---

> ### Author Response · Authors · 2025-03-17
> **Author response (Part 2)**
>
> ### 2. Inaccurate Experimental Approach: The authors claim that LLMs' poor performance on NLU tasks is due to their inability to capture bidirectional context and perform deeper semantic analysis. However, the rationale for choosing SFT and PPO to address this issue is insufficient. The authors fail to clarify why SFT and PPO enable the model to capture context and perform deep semantic analysis.
>
> We thank the reviewer for highlighting concerns regarding the rationale behind our experimental approach. To clarify, our primary goal is not to alter the causal attention mechanisms inherent in large language models (LLMs) to enhance their ability to capture bidirectional context or perform deep semantic analysis akin to encoder-only models. Instead, our aim is to explore effective fine-tuning methods—specifically proximal policy optimization (PPO)—to enhance the natural language understanding (NLU) performance of instruction-aligned LLMs, surpassing the limitations encountered with standard supervised fine-tuning (SFT).
> Our evaluations involve widely-used instruction-aligned models, including LLAMA2-7B-chat-hf, Qwen2.5-7B-Instruct, MPT-7B-chat, and LLAMA2-13B-chat-hf, which, despite their instruction alignment, demonstrate suboptimal zero-shot and few-shot performance on various NLU benchmarks compared to smaller, encoder-only models such as BERT. Across diverse NLU tasks—including Sentiment Analysis & Emotion Detection, Grammaticality & Linguistic Acceptability, Natural Language Inference, Question Answering & Reading Comprehension, Word Sense Disambiguation, and Commonsense & Reasoning—we consistently observe that PPO-based fine-tuning substantially improves model performance over SFT. For example, on the GLUE benchmark, PPO-fine-tuned models outperform SFT by an average of 6.3%.
>
> Thus, our experimental approach provides a robust and rigorous evaluation, demonstrating that PPO fine-tuning offers significant empirical improvements in performance for instruction-aligned LLMs on NLU tasks without relying on additional large-scale synthetic data or sophisticated reward models.

---

> ### Author Response · Authors · 2025-03-17
> **Author response (Part 3)**
>
> ### 3.Insufficient Analysis of Results: The paper merely describes the experimental results without providing a detailed theoretical explanation. For example, why does SFT underperform compared to PPO? How does PPO influence the model's parameter updates and semantic representations?
>
> We appreciate the reviewer’s valuable comment regarding the lack of a detailed theoretical explanation for why supervised fine-tuning (SFT) underperforms compared to proximal policy optimization (PPO), and how PPO influences parameter updates and semantic representations. While providing a thorough theoretical analysis is indeed important, it is beyond the scope of this current paper. We defer such theoretical exploration to future work.
>
> However, to further clarify and support our empirical findings, we performed two additional analyses, summarized below:
>
> #### **1. Additional Experiments on Cross-Domain Generalization**
>
> To examine instruction-following capabilities of fine-tuned LLMs in tasks differing from their original fine-tuning format, we conducted experiments with the LLAMA2-7B-chat-hf model fine-tuned on the SST-2 dataset (binary sentiment classification) and evaluated it on the Amazon Reviews task, which requires generating integer ratings from 1 to 5 based on review content.
>
> | Method    | Accuracy (%) | 95% Confidence Interval (CI)   |
> |-----------|--------------|--------------------------------|
> | Original  | 27.00        | [19.00, 36.02]                 |
> | SFT       | 0.00961      | [0.00, 0.03]                   |
> | PPO       | 39.35        | [38.39, 40.29]                 |
>
> - PPO Fine-Tuning demonstrates effective generalization and robust instruction-following, while clearly understanding the underlying task’s objective which is sentiment analysis in this case.
>
> - SFT Fine-Tuning, by contrast, is unable to generalize or adapt to differing input-output formats even though the task’s objective is the same which is sentiment analysis.
>
> Qualitative analysis further reveals that PPO effectively adheres to the instructions, consistently generating correctly formatted outputs along with analysis. Meanwhile, SFT consistently fails in format adaptation. We hypothesize this difference occurs because PPO fine-tuning, through the use of a clipping mechanism that limits deviations from the reference policy retains broader instruction-following and generalization capabilities. In contrast, SFT tends to force the model's parameters to learn a narrow, task-specific distribution, limiting flexibility and generalization. *We added this experiment to section 5.5 of the main manuscript.*
>
> #### **2. Language Modeling Evaluation on WikiText-2-Raw**
>
> We also assessed the impact of fine-tuning methods on the model’s general language modeling capabilities by evaluating perplexity (PPL) on the WikiText-2-Raw dataset, which follows a human-written natural language distribution.
>
> | Method    | Perplexity |
> |-----------|------------|
> | Original  | 6.939      |
> | PPO       | 6.966      |
> | SFT       | 7.384      |
>
> - The PPO-fine-tuned model maintains perplexity nearly identical to the original model, indicating negligible impact on its general language modeling capability.
>
> - In contrast, the SFT-fine-tuned model shows a notable increase in perplexity, reflecting deterioration in the model's ability to generalize beyond the training distribution.
>
> This further supports our conjecture that PPO effectively constrains updates via the clipping mechanism, thus maintaining the general semantic representations and broad language capabilities inherent in the original model. Meanwhile, SFT narrowly converges to the task-specific distribution, significantly impacting general language modeling performance. *This experiment has been added to the main manuscript in section 5.6.*

---

### Decision · Action_Editor_z7dX · 2025-04-17

**Recommendation:** Reject

**Comment:**

This paper studies using RL techniques (SFT and PPO) to enhance the NLU capabilities of LLMs. The authors demonstrate that PPO can significantly improve LLMs' NLU task performance (e.g., GLUE and SuperGLUE). The reviewers generally believed the results make sense, but also raised significant concerns regarding the technical depth (e.g., the paper applies RL techniques in a straightforward way) as well as the potentially outdated and insufficient evaluation results (e.g., evaluating mostly on old benchmarks like GLUE and SuperGLUE). Overall, the reviewers believed that the insights delivered by the paper are rather limited at this stage. Most of the reviewers recommended rejection in the final evaluation. We encourage the authors to prepare a major revision to address the identified weaknesses by the reviewers, especially the lack of insights beyond what is already known by the community (i.e., RL algorithms like PPO are effective in improving LLM capabilities). There are several ways that the paper could be significantly enhanced: (1) add more theoretical analysis to deepen the understanding of why PPO works for NLU; (2) add more evaluation on modern and challenging benchmarks to demonstrate when PPO is effective and when it is not; (3) propose new RL algorithms tailored for NLU.

**Audience:**

Most reviewers are concerned that the paper won't have sufficient audience at TMLR. Quoting the final recommendation from one reviewer:
"TMLR readers are likely already familiar with techniques like LoRA and PPO, and the submission does not offer much new insight into why PPO performs better in this setup or what conceptual takeaways we gain beyond what is already known. As such, it's not clear what this paper adds for the TMLR audience."

**Claims And Evidence:**

Yes

**Resubmission Of Major Revision:**

The authors may consider submitting a major revision at a later time.